# Cloud Workload and Data Center Analytical Modeling and Optimization Using Deep Machine Learning

**Tariq Daradkeh** *,† and **Anjali Agarwal** †

Electrical and Computer Engineering Department, Concordia University, Montreal, QC H3G 2W1, Canada
* Correspondence: tariqghaleb.daradkeh@concordia.ca; Tel.: +1-514-912-8382
† These authors contributed equally to this work.

**Abstract:** Predicting workload demands can help to achieve elastic scaling by optimizing data center configuration, such that increasing/decreasing data center resources provides an accurate and efficient configuration. Predicting workload and optimizing data center resource configuration are two challenging tasks. In this work, we investigate workload and data center modeling to help in predicting workload and data center operation that is used as an experimental environment to evaluate optimized elastic scaling for real data center traces. Three methods of machine learning are used and compared with an analytical approach to model the workload and data center actions. Our approach is to use an analytical model as a predictor to evaluate and test the optimization solution set and find the best configuration and scaling actions before applying it to the real data center. The results show that machine learning with an analytical approach can help to find the best prediction values of workload demands and evaluate the scaling and resource capacity required to be provisioned. Machine learning is used to find the optimal configuration and to solve the elasticity scaling boundary values. Machine learning helps in optimization by reducing elastic scaling violation and configuration time and by categorizing resource configuration with respect to scaling capacity values. The results show that the configuration cost and time are minimized by the best provisioning actions.

**Keywords:** machine learning; cloud elasticity; optimization; workload prediction; data center resource configuration; elastic scaling



## 1. Introduction

The ultimate goal of cloud infrastructure resource management is to achieve a tradeoff between two contradicting factors: reducing data center operation with minimum configuration and running cost and ensuring Service Level Agreement (SLA) by maintaining the Quality of Service (QoS). Cost reduction requires minimizing allocated resources to maximize resource utilization. On the other hand, service performance requires increasing cloud resources to respect SLA and guarantee QoS. The cloud model achieves that by elastic scaling of cloud resources dynamically, which is one of the components of cloud definition in NIST [1]. Elastic scaling is performed either by increasing (to ensure QoS) or by decreasing (to ensure minimum cost) resources, such that workload demands are accommodated each time, as performed in the "pay as you go" elasticity model [2]. A cloud management system is an integration of service components that works on maintaining cloud resources and services to respect SLA. Cloud manager actions rely on the system status for both cloud resources usage and configuration time (scheduling and orchestration), and cloud application performance and behavior (workload demands). The cloud manager must include the following modules: (1) monitoring modules that keep track of resources and application status, (2) the job scheduler module works on assigning tasks to proper computing resources, (3) the orchestrator module works on physical and platform resources management by switching on/off physical/virtual servers or enabling/disabling

hardware modules or services (hardware reconfiguration), and (4) a cloud general manager, which correlates the module's operations and tasks. The general cloud manager must include the kinds of optimization modules that work with the scheduler to achieve the best resource utilization and minimum cost. Classical optimization methods, by formulating cost functions according to certain goals, such as bin pack, particle swarm and best-fit algorithms, are non-polynomial NP-hard problems that cost time to solve and some time constraint relaxations. Finding a generalized form for a cloud system will help from different perspectives: First, it will help in workload and system behavior prediction, which can help give the optimizer the time to find the best scheduling and orchestration setup. Second, it will avoid tedious resource reconfiguration that might iterate multiple times just by following the demands, which stresses the hardware and might cause failure. By having data center and workload demand models, the general manager can define the configuration time, scaling direction, resource capacity and reconfiguration set. This will help avoid any SLA violations and achieve elastic scaling.

The cloud management system must find the optimal solution for elasticity in scaling cloud data center resources, and this solution is required in the Infrastructure as a Service (IaaS) cloud layer. The elasticity feature requires a deep understanding of two components; (i) the workload and (ii) the data center's resource capability and configuration. The workload consists of the demand, i.e., the amount of resources that must be allocated to perform the job in a specific time unit, and their characteristics, i.e., demand shape, velocity, volume, frequency and duration. A data center's capability includes a list of physical machines (PMs) and their processing power, memory, storage, network capacity, hardware and virtualization technologies, hypervisor models and virtualization level (virtual machines (VMs) or containers). Configuration deals with start-up or spinning time [3], applications running on the VMs or containers, service component dependency, such as affinity, security, availability, and applied running protocols.

Most of the cloud resource management models are focused on optimizing resource utilization while ensuring QoS by performing VM scheduling (consolidation, replacement, load balancing and migration) within a single-cloud data center or via multiple geographically distributed data centers. The intuitive idea for VM scheduling is mapping VM to PM as a bin packing problem to optimize resource usage for automatic, elastic scaling. This problem is considered non-deterministic polynomial-time hardness (NP-hard), as discussed in [4]. Considering this, a near-optimal solution can be proposed as an optimization problem subject to one or multiple objective constraints. Many works have been undertaken for elastic action in general that are based on optimization, either directly or indirectly. Some of the cloud management models used prediction methods based on statistical time series analysis and probability stochastic theory to anticipate different parameters or to model the system, i.e., predict or model workload, scaling direction, user behavior and elastic resource scaling.

Machine learning (ML) has been used in cloud management systems for different purposes. Unsupervised learning methods, such as clustering and reinforcement learning, are used for workload classification and resource scaling. Supervised learning, such as classification and regression, is used for workload prediction and data center resource configuration. Deep machine learning is an intensive method for problem modeling and solving, which can be considered as a pluggable framework to solve complex and high-dimensionality problems. ML, in general, can be used in the following cases of optimization problems: non-convex cost function, nonlinear relation, high dimensionality factors and complex and dynamic environments [5,6]. In many data-mining applications, deep machine learning provides a more robust solution as it works by stacking many layers to solve nonlinear and complex problems. Using machine learning and/or deep learning in a cloud management system to solve the scaling goals is not new. However, using it as a holistic optimization model for elasticity modeling, comprising both workload and data center configurations, is a novel idea. To the best of our knowledge, this approach for elasticity modeling has not been used before.

Our contributions are summarized as follows:

- Logs trace analysis for Google and Alibaba cloud data centers, showing elastic proportioning.
- Workload analytical modeling and evaluation to be used for forecasting and evaluation.
- Data center configuration setup and operation modeling to evaluate resource configuration and scaling time.
- Elastic scaling evaluation using boundary limits to find scaling violation and matching ratios.
- Optimizing elastic scaling using deep machine learning.

The rest of the paper is organized as follows. Section 2 presents the related work. Google and Alibaba trace descriptions are presented in Section 3. Workload and data center modeling are given in Section 4. Section 5 shows the elasticity operation parameters, and Section 6 presents the elastic scaling optimization using deep machine learning. The conclusion is given in Section 7.

## 2. Related Work

In most of the cloud elasticity scaling and resources management models, optimization is the key to achieving the goal. A classic optimization method using a cost function subjected to multiple objectives is used in [7–12]. A QoS cross-entropy stochastic modeling of workload scheduling algorithm called CESS is proposed in [7] for SaaS layer demand. The authors of this work focused on multiple optimization objectives for QoS constraints, i.e., for job time constraints (timeliness), reliability of the data center to complete the jobs and security in running the job. In [8], the authors proposed a framework called UPPAAL-SMC to evaluate and optimize cloud resource allocation using statistical and unsupervised learning methods. Their proposed framework came to quantify cloud resource allocation capabilities and to optimize provisioned cloud resources usage, considering the quality of service (QoS) in the service level agreement (SLA). The authors of this work used Statistical Model Checking (SMC), i.e., sequential hypothesis testing to measure QoS probability, and custom supervised learning to optimize Resource Allocation Instances (RAI). Optimized resource allocation using Adaptive and Fuzzy logic Resources Management (AFRM) is proposed in [9]. The authors used a scale parameter based on workload change instead of a constant coefficient in the proposed adaptive rule-based algorithm. Fuzzy logic is integrated with the algorithm of resource demand prediction. In general, most of the optimization methods are subjective to a limited number of case scenarios, which can influence resource allocation accuracy and utilization. However, we used deep machine learning models as dynamic cost function formulators that adapt to system characteristics changes.

The bin packing approach is proposed in [13,14] to assign VM to a physical machine for auto-scaling and optimum hardware utilization. The authors of [13] focused on automatic scaling using a shadow algorithm for VM to PM mapping, i.e., for solving the bin packing problem. The optimization cost function is considered as the minimum number of PMs to be used in each configuration time using the shadowing queuing model. In [14], the same dilemma was tackled for scaling using a color set algorithm by optimizing three cost functions to achieve minimum replacement, lower cost and energy savings. In [15], the researchers compared the bin packing algorithm and Genetic Algorithm (GA) for cloud resource allocation optimization. The goal of this work was to make an elastic provision that balances QoS and resource cost. Hotspot elimination in cloud data centers using the stochastic optimization method are proposed in [16,17]. Workload analysis and modeling using stochastic random variables to reduce the VM migration overhead is proposed in [16]. A load balancing algorithm is proposed for VM migration, which answers three questions: when to migrate, where to send the VM and which VM to migrate. The objective function is to minimize hotspot cases and VM migration costs based on workload change. In [17], a Tranche Markov Prediction algorithm for cloud load balancing is proposed to minimize the VM migration cost and remove hotspots in the cloud data center. SLA is considered in both works as a goal to be achieved.

An admission test of services to achieve optimal resource allocation in an elastic manner was proposed by Konstanteli et al. in [10,11]. In [10], an optimization model is proposed to find the best allocation that can achieve efficient performance, minimum cost and affinity of service VMs, using the General Algebraic Modeling System (GAMS) to model the problem and solve it under real test cases. On the other hand, in [11], the authors focused on the probabilistic optimization method to achieve cloud elastic service admission subjected to the same goals (efficiency, cost and affinity). A heuristic solving model is integrated into the optimization method to solve the MINLP non-convex problem and reduce its complexity. A capacity planning and admission control model is proposed in [12] to maximize revenue and optimize capacity allocation. The authors proposed a stochastic dynamic programming algorithm with two heuristic objectives that can tradeoff between computation complexity and the optimal solution. Workload modeling, evaluation and forecasting are investigated in [7,16,18–20]. Both [7,16] used a stochastic queuing scheme to model cloud data center workload as a time variable and dynamic behavior. A Platform as a Service (PaaS) elastic model (AutoElastic) is proposed in [18] for HPC clusters in a cloud environment. The authors of this work contribute in two ways: (a) making an aging technique for cloud elasticity actions instead of using static threshold values on VM provisioning and de-provisioning, and (b) applying an asynchronous approach in VM management (create/delete/modify VM instances) that can achieve parallel execution in cloud management actions. This allows parallelization in VM allocation for parallel applications. Our concern in [18] is the asynchronous model might cause conflicts in management actions and may violate elastic conditions. This is because the asynchronous model type does not consider all application performance parameters concerning resource provisioned values.

Prediction, consolidation and migration modules are proposed in [19]. In this work, the authors used the ARIMA forecasting method to predict the workload and estimate the allocated cloud storage resources. A combination is proposed between the prediction of resource demands (Resource Demand Estimator) and server consolidation to minimize the data migration time. The ARIMA forecasting method is used to predict the workload in [20] for SaaS cloud providers. Using the proposed predictive resource allocation method, the authors analyzed the system accuracy by an enhancement factor in QoS conditions and resource utilization. An auction-based provisioning model for cloud IaaS instead of a "pay as you go" scheme (elastic scheme) is proposed in [21]. The authors of this work proposed a geographically distributed cloud data center to scale IaaS resources. Their work is based on a classic optimization method with a cost function subjected to maximizing the revenue of the cloud owner with respect to the cost of provisioned resources (resource allocation). Zhang et al. in [18,22] worked on an auction-based resource allocation model for efficient resource allocation and the best cost and pricing scheme. In [22], an auction-based resource provisioning via geographical data centers is proposed. The authors of this work proposed a randomized auction method for dynamic VM provision, using smoothed analysis in their algorithm to reduce its NP-hard problem complexity of resource allocation. In [18], the researchers proposed efficient VM provisioning and pricing using an online auction for VM scaling. With this auction, model users can continually bid for needed VM using both scaling methods (vertical scale-up and horizontal scale-out). The limitation of [18,22] is that the auction model might reduce the cost on the cloud consumer side but does not utilize resource allocation efficiently, especially in a geographical data center. It will waste a small amount of available resources for each data center member, which can be consolidated and rebalanced to accommodate a larger size of jobs.

Cloud elasticity has been modeled and analyzed in a quantitative approach in [23] for predicting elasticity in a cloud environment. The author proposed a probabilistic approach of computing provisioned resources matching the demand workload by modeling elasticity as a queuing system using a continuous Markov chain (CTMC). The parameters considered in the queuing model are task arrival rate, service rate, VM spanning time (VM start-up and service running time) and VM shutdown. The quantitative predicted elastic attributes

are task response time, system throughput (tasks finished), QoS (performance), number of provisioned VMs, cost and scaling. A tiny mapping between workload and cloud resources configuration is proposed in both works [24,25], where a tailored matching algorithm between cloud computing machine list (number of machines in each infrastructure type) with respect to workload need is proposed in [24]. The authors of this work focused on a many-task computing (MTC) workload type using linear optimization to increase the number of tasks completed per time unit. The optimization proposed has multiple objectives of minimizing cost, energy and machine failure list and maximizing the number of tasks finished per time unit. In [25], the authors focused on minimizing the SLA violation rate in the context of workload execution time. The workload types considered in this work were diverse from multiple kinds of applications. The compound parameters to be satisfied are cost, task execution time, reliability, latency and availability.

An elastic virtual machine provision prediction model based on workload history is proposed in [26]. The authors compared three regression models, i.e., the ARIMA statistical method, neural network and Support Vector Machine, for forecasting provisioning, and they used the Kalman filter method for raw data pre-processing. In [27], a neural network framework was proposed for the accurate prediction of cloud data center server workload (PRACTISE). The authors compared their module PRACTISE with the ARIMA forecasting method and baseline neural network and showed that the proposed method achieved better estimation for the provisioning of resources. A hybrid prediction method for cloud resource usage of container workload using ARIMA and Triple Exponential Smoothing is proposed in [28]. The authors focused on optimizing the Docker container's CPU resource usage based on predicted demands. A Recurrent Neural Network Long Short-Term Memory (RNN-LSTM) prediction method was used in [29] to estimate the cloud workload on data center servers to allocate and deallocate resources. The authors compared their approach with the ARIMA statistical forecasting model and showed that LSTM has a better prediction accuracy. The Multilayer Perceptron (MLP) and LSTM prediction methods are used to forecast cloud workload in [30]. The authors compare their work with ARIMA, and the result indicates that the proposed method has higher accuracy. Reinforcement Learning (RL) has been used in cloud elastic scaling in multiple works [31–34]. Workload load prediction and reinforcement learning are used for cloud automatic scaling in [31]. The SARSA algorithm is used to predict future reward estimates of the 1998 World Cup dataset of the access traffic to test the system, where the Q-function considers the predicted values and resource usage in resource scaling. The authors of [32,33] used deep reinforcement learning methods for cloud elastic resource provisioning. In [32], the proposed method is called Deep Elastic Resource Provisioning (DERP). The deep reinforcement learning algorithm works by combining deep neural networks and Q-learning to rescale cluster resources. The workload is synthesized mathematically and the proposed system is tested under two case scenarios, where a neural network is used as a function approximator, and Q-learning logic is used for determining the next action. Moreover, in [33], the authors try to find the balance between the exploration and exploitation of elastic schedule GPU resources. An open-source library, ElegantRL, is developed to utilize and exploit cloud GPU resources and achieve elastic scaling. A NASDAQ-100 dataset is used as an experimental test for workload simulation. A reinforcement learning-based controller to respond to the complex workload demand elastically using a set of states is proposed in [34]. The authors used queuing theory to model the system and meanwhile used unsupervised ML reinforcement learning to manage the resources by the reading system and application states and to formulate the Q-logic in order to manage resources. This model focused on rewarding factor values for long-term execution considering the ultimate goal of elastic provisioning of resources while respecting application performance.

A full taxonomy of the related work is presented in Figure 1, with a summary of the methodologies used in resource scaling and cloud management systems.

Overall, many works have discussed general cloud management and optimization models, considering different cloud services and optimization goals. In [35–37], a cloud

management platform with self-adaptation and model management was used as software for a service cloud layer. The authors of [35] introduced a model-driven-based system-model management operation (MMO) to reflect the software model on cloud resource infrastructure. Whereas in [36], the authors produced a full resource management framework based on a feedback loop to allocate resources. In [36], an application-oriented model management method for dispatching and controlling cloud resources is shown. The contributions from [38,39] are focused on cloud workload optimization to reduce power consumption and enhance performance. In [38], the authors focused on managing resource orchestration and job scheduling using application-type analysis. Where in [39], the authors used Particle Swarm Optimization methods to schedule job submissions.

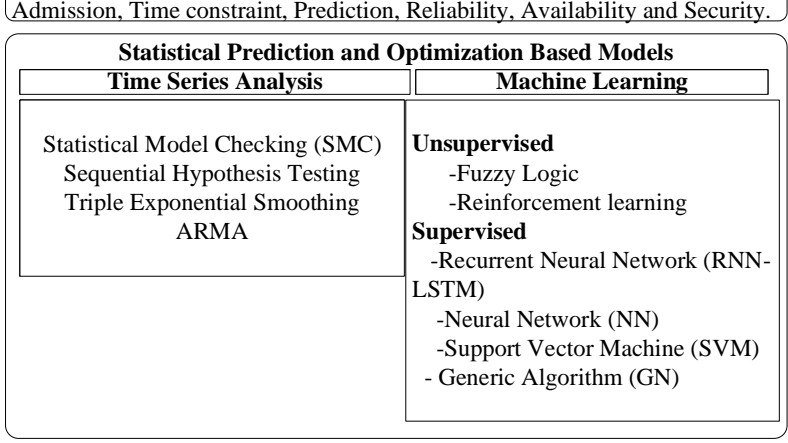

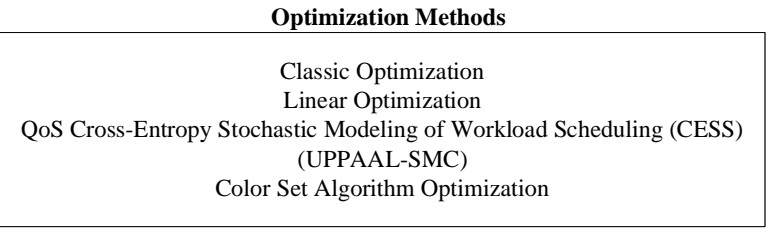

**Figure 1.** Review of Elastic Resources Scaling Methods and Taxonomy.

## 3. Google and Alibaba Traces

The Google data center log trace has been investigated, in detail, in [40], where the workload is assigned as a set of jobs, and each job has a set of tasks. Each task has a CPU, RAM and Disk I/O resource demand (*RD*). There are nine states that tasks can go through during the execution where resources scale; either scale out to tasks in Submit, Schedule, Pending and Running states or scale in when tasks are in Evict, Fail, Finish, Kill and Lost states. Resource provision (*PR*) is achieved by adding, modifying or removing the physical server. Server resources normalize to the maximum resource value, where the four machine types that are normalized for maximum resources are 1.0, 0.75, 0.5 and 0.25 resource units.

The Google workload demand and data center resource provisioning relationships are shown in Figures 2 and 3.

In elastic scaling, there are three cases of provisioning resources, releasing too many resources (much more resources than is required), called over-provisioning ($OP$), matching the demand condition as in SLA, called fit-provisioning ($FP$), or giving fewer resources than needed, called under-provisioning ($UP$). It is clear that in Figure 2, an under-provisioning case happens, and in Figure 3, an over-provisioning case happens after 600 min. The fit-provisioning case occurs in both figures with limited time intervals. These two figures show the real workload demand and how the data center reacts to it. In both workload types, there are violations of either over-provisioning or under-provisioning that can be seen visually.

In practical scaling, it is very important to define the boundaries of resource scaling, which defines over/under/fit-provisioning states. The boundaries of the Upper Bound ($UB$) and Lower Bound ($LB$) define the range of the fit state. Usually, the Lower Bound is equal to the exact demand ($LB = RD$), and the Upper Bound is greater than the demand with a certain base value $B$ where ($UB > RD + B$). Under-provisioning happens when the provisioned resources are lower than the requested demand ($UP = PR < RD$). Fit-provisioning ($FP$) is achieved when the provisioned resources are between the $LB$ and the $UB$ ($FP = (RD + B) \geq PR \geq RD$). Over-provisioning ($OP$) is defined when the provisioned resources are more than the sum of demand and boundary value ($OP = PR > (RD + B)$). Defining the boundary value $B$ is a challenging task because the workload demand shape is non-deterministic. For Google trace, it is shown statistically that $B = RD \times 0.2$ would be fair for resource utilization and system performance in sample 1, Figure 2, since the demand increases non-continuously. However, in the case of a gradual demand increase, as in sample 2, Figure 3, the resources provisioned become very large, and resource wastage increases dramatically.

A general gradual increase in the requested demand is depicted in Figure 4, where the Upper Bound represents the sum of the demand and boundary ($RD + B$), and the Lower Bound is equal to the actual demand. Note that in Figure 4, the curve for the Lower Bound is the same as the demand curve. The best provisioning value is the provision line that has a value close to the demand and a variable base value, $B$, as it is clear that the demand was small in the beginning and the boundary value was $B = 0.5 \times RD$; however, with higher demand values $B = 0.01 \times RD$. The value of $B$ must be reasonable with respect to the demanded resources provisioned, such that it is as close as possible to the demand. On the other hand, fixing the $B$ value as a small constant factor will cause many provisioning problems because the configuration time and resource preparation has a relationship with the total demand. The value of $B$ must dynamically change as the demand increases. In Figure 4, the machine's contribution in the provisioning process for a static $B$ is derived from the real trace cases, which means the amount of resources released must achieve the requested demand with good performance. For provisioning with an adaptive $B$, a machine list is dynamically rearranged, and the best resources are chosen. A rational derivation for $B$ values is to correlate the minimum and maximum demand values during a time interval ($RD_{min}$, $RD_{max}$) with respect to the maximum and minimum provisioned values ($PR_{min} and PR_{max}$). The relationship is derived by finding the minimum value of the normalized averaged of the minimum demand and what was provisioned and the normalized average of the maximum demand and what was provisioned during the time window ($T_w$). The final boundary value is described in Equation (1) for each time window.

$$B = \min\left\{ \frac{PR_{max} + RD_{max}}{T_w \times RD_{max}}, \frac{PR_{min} + RD_{min}}{T_w \times RD_{min}} \right\} \times RD. \tag{1}$$

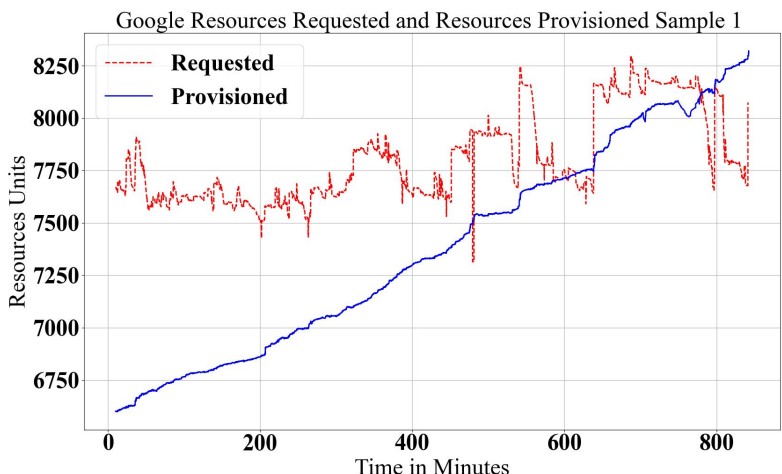

**Figure 2.** Google data center Provisioned and Requested Workloads, Sample 1.

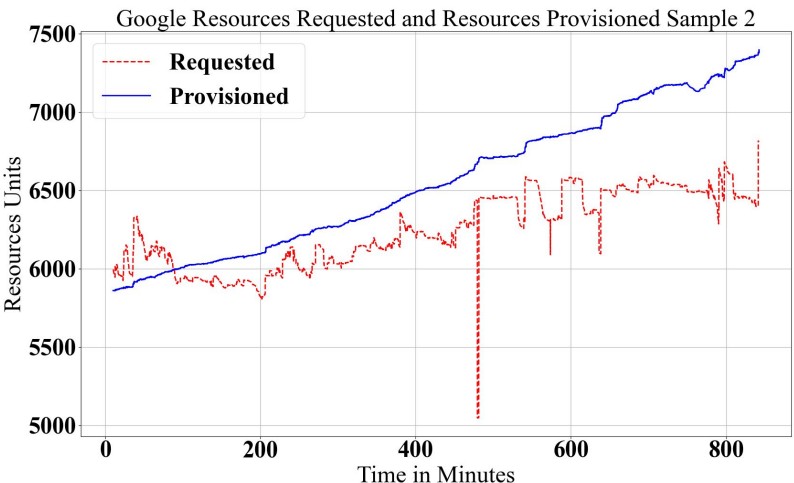

**Figure 3.** Google data center Provisioned and Requested Workloads, Sample 2.

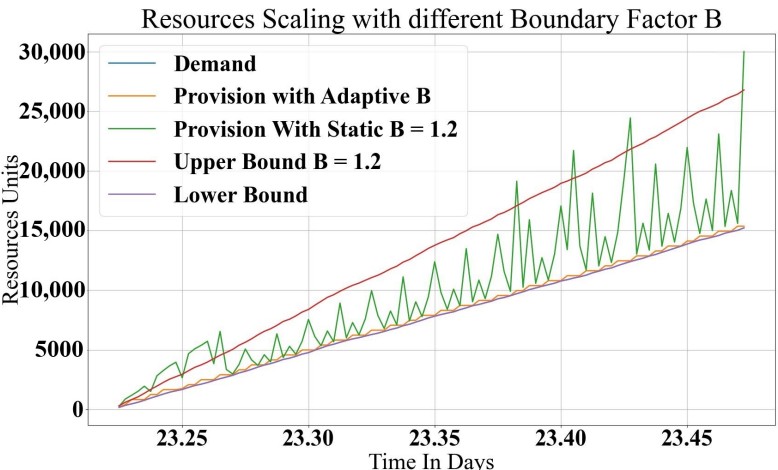

**Figure 4.** Google data center with Constant and Adaptive B Ratios and Gradually Increasing Resources.

The Alibaba data center log trace [41] has similar characteristics to Google trace, where the workload is submitted in a set of jobs. Each job has a set of tasks, and in each task, there are a set of instances. An instance is a copy of the task operation with different input data. A task at Alibaba has six states, where Ready, Waiting and Running need resources (scale out),

and Terminated, Failed and Canceled do not need resources (scale in). For each instance, there is an average usage of RAM and CPU. Instances are used as the demand reference, and the states indicate the scaling direction because each instance is run in the task. In the schema, the workload is found by multiplying the number of instances by the CPU usage value for CPU demand and RAM usage value for RAM demand during the time window $T_w$. Workload demand is found based on instance (Ready, Waiting, Running) states means that resources are requested, and other states mean resources are to be released. Again, values must be normalized, the RAM demand is already normalized, but the CPU demand is not, which is normalized during data preparation. For provisioning of the resources, the machine has three states (add, soft error, hard error). The adding machine is used to scale out. For scale in, it is not specified, but it is considered as soft or hard errors. In Alibaba trace, more Terminated states are shown, which gives a negative value because the trace is cut into small time durations of 12 hours and does not show the mapped start time and ID for the terminated instances. It is fixed by adding a biased value (negative minimum value) for both demand and provisioning, based on which scaling is found. Figures 5 and 6 depict the Alibaba data center scaling in reaction to workload demand.

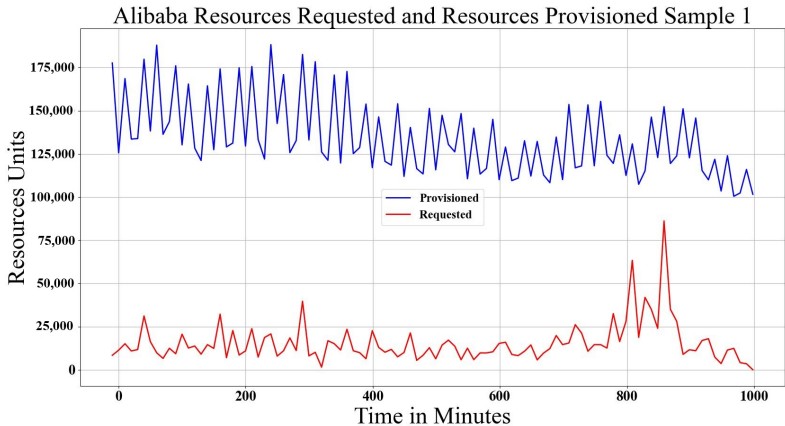

**Figure 5.** Alibaba data center Provisioned and Requested Resources, Sample 1.

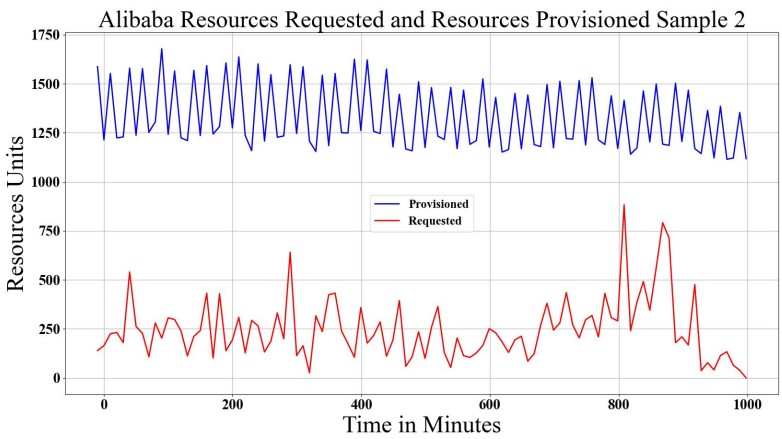

**Figure 6.** Alibaba data center Provisioned and Requested Resources, Sample 2.

Table 1 summarizes the mathematical notations and their meaning that have been used in this section. It is important to mention that the trace of workload demand and provisioned resources in Google and Alibaba are totally different, but, in general, they are similar to an abstract view of the attributes of scaling and provisioning. In the next section, Google traces are modeled analytically in comparison with Alibaba.

**Table 1.** Key Symbols Used in the Analysis and their Definitions.

| Symbol | Definition |
| --- | --- |
| | Google and Alibaba Traces Section Symbols |
| $RD$ | workload resources demand |
| $PR$ | data center resource provision |
| $OP$ | over-provisioning case, $(OP = PR > (RD + B))$ |
| $FP$ | fit-provisioning, $(FP = (RD + B) \geq PR > RD)$ |
| $UP$ | under-provisioning $(UP = PR \leq RD)$ |
| $B$ | base value |
| $LB$ | Lower Bound, $(LB = RD)$ |
| $UB$ | Upper Bound, $(UB > RD + B)$ |
| $T_w$ | time window interval |

## 4. Workload and Data Center Modeling

A complete model for data center behavior and workload demand is presented in this section using two methods. The first method uses analytical and statistical probability to describe data center provisioning and workload demands. The second method uses deep machine learning by correlating the input and output behavior of the system. Figure 7 shows the general architecture of workload demands and the data center with its internal modules. Workload demands are described by bunches of jobs that include tasks submitted to the data center queue (the Submit Tasks ellipse in Figure 7). Data center resources (Resources Machine List) are configured on the fly to match the requested demands in an elastic way by increasing or decreasing the provisioned resources. The system receives submitted jobs/tasks as a script file descriptor, which specifies the needed resources, similar to HPC job submission. The submitted tasks module acts as a proxy to communicate with the resources scheduler that masters the resource reconfiguration and assigns tasks to computational nodes with proper reconfiguration. The machine list is updated by the data center capacity planning, which is the total resource capability. The data center is scaled all the time by adding new servers or removing failed machines. The operation configuration works with the scaling action to determine the direction and reflect it in the machine classes. The machine list is also updated by the setup configuration, which works to accommodate new tasks and by the terminate configuration, which works to release the resources from terminated or failed tasks. The failed tasks will be resubmitted by en-queuing them in the Submit Tasks. The two models must represent the whole architecture accurately that will be used to validate and test our proposed optimized machine learning elastic model.

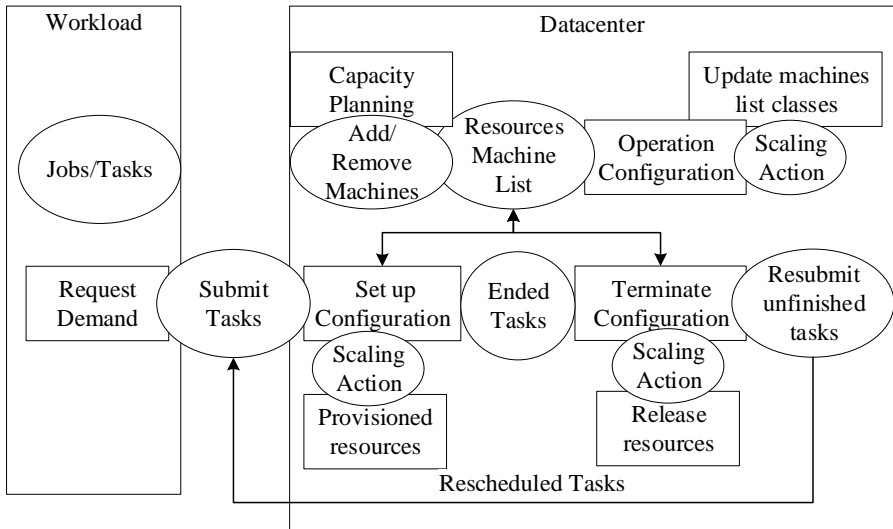

**Figure 7.** Architecture of the data center and workload models.

### 4.1. Analytical Modeling Approach

Workload demand and data center resource provisioning are modeled using statistical probability and stochastic random variables. The workload model section focuses on modeling the demands using statistical analysis and probability random variable theory, whereas, for data center modeling, a queuing theory is used based on a stochastic model.

#### 4.1.1. Workload Modeling

A full model of workload demand prediction is proposed in our previous work [42]. This work is a continuation of the modeling operation, which will be used for the validation and testing of the optimized machine learning elastic scaling.

The proposed analytical model describes the demands of the workload generated by clients' applications in all cloud service layers (AaaS, PaaS and IaaS), where the overall demands will be reflected in data center infrastructure resources (the IaaS layer). The infrastructure as a service layer provides computing resources as physical and virtual resources units, such as PM, RAM, CPU, Network, Storage, VM and Container. The Google cloud data center and workload demands were investigated in our previous work [40]; we formulated the analytical model by mapping out its features using statistical analysis. In a time window $T_w$, there are $(J_s)$ submitted jobs and $(T_s)$ submitted tasks that construct the $(W)$ cloud workload. Workload demands request cloud resources during a time that is counted by time window segments $T_w$ that affects the demands $J_s$ and $T_s$ by increasing or decreasing cloud resources. There are tasks that will end $T_e$, of which some will end and finish successfully, $(T_{cf})$ and others will terminate $(T_{ne})$ due to failure or eviction or being killed or getting lost.

Submitted tasks are served by cloud resources that are provisioned in response to demands and released once these jobs $J_e$ and tasks $T_e$ end. For Alibaba, the equivalent of a task is an instance. Each submitted job $j$ includes a batch of tasks $(T_j)$, where one task requires a vector of infrastructure resource demands $(R)$. Provisioned and de-provisioned data center resources are counted in resource unit $R_{ij}$ of task $i$ in job $j$. For tasks that have ended, $RE_i$ is the released resource vector of integer index $i$.

Equation (2), depicts the requested resources $R_t$ in $T_w$ duration at time $t$, with a number of submitted tasks $T_{jt}$ in $J_{st}$ submitted jobs.

$$R_t = \sum_{j=1}^{J_{st}} \sum_{i=1}^{T_{jt}} R_{ij}. \tag{2}$$

The total released resources $RE_t$ depends on ended tasks $T_{et}$ during time window $t$, as Equation (3) depicts.

$$RE_t = \sum_{i=1}^{T_{et}} RE_i. \tag{3}$$

$T_{ne}$ are the ended tasks that are not completed but terminated due to being failed, evicted, killed or lost. These tasks will be resubmitted, $RR_t$, in the next time window, as Equation (4) depicts.

$$RR_t = \sum_{i=1}^{T_{net}} RE_i. \tag{4}$$

A data center's resources demanded $(RD_t)$ by workload at time $t$ during time window $T_w$ is given by Equation (5).

$$RD_t = R_t + RR_{t-1} - RE_t. \tag{5}$$

Equation (6) represents the total workload demand $W_t$ at the current time $CT$, starting from time $t = 1$ as a count of time window $T_w$. For Alibaba, the calculations are similar,

with three parameters; the number of instances in the task, the number of tasks in the job and the number of jobs multiplied by instances of resource demand.

$$W_t = \sum_{t=1}^{CT} RD_t. \tag{6}$$

Using the above equations, the resource demands are found, and the following parameters $J_s, T_s, T_j, R_{ij}, T_e, T_{ne}$ and $RE_i$ are extracted as stochastic probability events. However, these values are time-variant as stochastic random variables. The workload demands $W$ is found using the aforementioned equations using $J_s$ arriving jobs, $T_j$ number of tasks per job, $T_s$ total submitted tasks, $R_{ij}$ requested resources per task, $T_{cf}$ ended finished tasks, and $T_{ne}$ the ended nonfinished tasks.

A normal random variable (Gaussian distribution) is used to model the number of submitted jobs $j$, the total submitted jobs $J_{st}$ and the number of submitted tasks $i$ of total submitted tasks $T_{st}$ probabilities during time interval $T_w$ as depicted in Equation (7). In Google traces, it is clear the demand has a bell shape curve during time intervals with some skew to the left or right, as in Figures 2 and 3. As we investigated in our previous work [40], the number of submitted jobs and their tasks during one time window are counted as workload characteristics for the scaling time slot. However, in Alibaba trace, it is clear that the Gaussian random distribution is not the best representation. The shape of the trace is more triangle, which means the best distribution is Triangular distribution, as can be seen in Figures 5 and 6.

$$P(J_{st}(j)) = \frac{1}{\sqrt{2\pi}\sigma_j} e^{\frac{-(j-\mu_j)^2}{2\sigma_j^2}},$$

$$P(T_{st}(i)) = \frac{1}{\sqrt{2\pi}\sigma_i} e^{\frac{-(i-\mu_i)^2}{2\sigma_i^2}}. \tag{7}$$

The sample mean of the jobs and tasks $\mu = \bar{x}$ and variance $\sigma^2 = s^2$ are found statistically, as given by Equation (8). The values of the mean and variance are updated in each time window to be used in the next window prediction. This makes the system adaptive to changes. The vector variables $J_{st}(j)$ and $T_{st}(i)$ specify submitted jobs and tasks, and their numbers are referenced by a continuous index. The number of submitted jobs $J_{st}$ and number of submitted tasks $T_{jt}$ are found from the length of their vectors. The total number of jobs $J = \sum_{t=1}^{CT} J_{st}$ and the total number of tasks $T = \sum_{t=1}^{CT} T_{st}$ are found statistically from the trace log and maintained cumulatively.

$$\bar{x}_J = \frac{1}{CT}\sum_{t=1}^{CT} J_{st}, \; s_J^2 = \frac{\sum_{t=1}^{CT}(J_{st} - \bar{x}_J)^2}{J-1},$$

$$\bar{x}_T = \frac{1}{CT}\sum_{t=1}^{CT} T_{st}, \; s_T^2 = \frac{\sum_{t=1}^{CT}(T_{st} - \bar{x}_T)^2}{T-1}. \tag{8}$$

In task $i$ of job $j$, $R_{ij}$ is the requested resources vector modeled by a uniform probability distribution. The requested granular resource units are similar because the minimum demand resource vector units are fixed, such as for database query or response web servers. The random variable expected value $E(X)$ is used in Equation (9) to find the resource vector for submitted task $R_{ij}$ and ended task $RE_i$, which is equal to one resource unit if uniformly distributed.

$$RE_i = R_{ij} = E(R_{ij}) = \frac{(R_{ij(t+T_w)} + R_{ijt})}{T_w}. \tag{9}$$

At each time window, the number of ended tasks is a Gamma distribution model, as depicted in Equation (10). The events that happened follow a Gamma distribution, where the number of ended tasks out of the total submitted tasks needs to be found. Tasks that ended within the time constraint and occurrences are modeled using a Gamma distribution with two parameters, $\lambda$ and $r$, called scale and shape, respectively. In a Gamma distribution, $\lambda$ is an important attribute, which, when found statistically, can be used to define the mean $\mu_\Gamma = \frac{r}{\lambda}$ of the population, and $r$ is the number of events (in our case, $r = T_{et}$ out of $T_{st}$). The similarity in this case between Google tasks and Alibaba instances is the same type of distribution.

$$\Gamma(T_{et}) = \int_{t=1}^{T_w} (T_{st})^{(T_{et}-1)} e^{-(T_{st})} dT_{st}, \quad T_{st} > 0,$$
$$P(T_{et}) = \frac{\lambda^{T_{et}} T_{st}^{(T_{et}-1)} e^{(-\lambda T_{st})}}{\Gamma(T_{et})}, \quad T_{st} > 0. \tag{10}$$

There are two cases for ended tasks $T_{et}$. Type one is the tasks that are completed and finished, normally $T_{cft}$, where the allocated resources are permanently released. Type two ($T_{net}$) is the tasks that end before completion and are not terminated normally. These tasks will be resubmitted $T_{re}$ to the next time window. A hypergeometric distribution is used to describe $T_{net}$ tasks that are not finished normally as a probability ratio of failure to success tasks in the submitted tasks batch ($T_{ret} = T_{s(t-1)} - T_{cf(t-1)}$) without any order, as Equation (11) depicts. Here, $T_{ret}$ represents the total number of resubmitted tasks in a specific time window.

$$P(T_{net}) = \frac{\binom{T_{et}}{T_{net}}\binom{T_{ret}}{T_{cft}}}{\binom{T_{st}+T_{re(t-1)}}{T_{et}}}, \quad T_{net} \leq T_{et}. \tag{11}$$

The Poisson process is used to model the number of submitted jobs and tasks with the average arriving rate $\mu_j$ multiplied by time window $T_w$, multiplied by the probability cumulative distribution function CDF $P(J_{st}(j) \leq j)$. The job and task arriving process is depicted in Equation (12). The same relation applies to Alibaba traces, where the Poisson process and exponential random variable describe the submitted and ended instances.

$$J_{st}(j) = \mu_j \times T_w \times P(J_{st}(j) \leq j),$$
$$T_{st}(i) = \mu_i \times T_w \times P(T_{st}(i) \leq i). \tag{12}$$

The exponential random variable process is used to model the number of ended tasks $T_{et}$, where the exponential random is a special case in a Gamma distribution when $r = 1$. The expected value of the number of non0finished tasks $T_{net}$ is applied to a hypergeometric distribution $E(T_{net}) = \frac{T_{et}}{T_{st}}$ as Equation (13) describes the ended task states (ended tasks that complete $T_{cft}$, and tasks ended without completion $T_{net}$).

$$T_{et} = \mu_\Gamma \times \Gamma(T_{et}) \times P(T_{et}) \times T_w,$$
$$T_{net} = T_{et} \times P(T_{net}) \times E(T_{net}) \times T_w, \tag{13}$$
$$T_{cft} = T_{et} - T_{net}.$$

Algorithm 1 describes the workload modeling using the previously defined equations. Table 2 summarizes the workload and data center mathematical model notations.

---

**Algorithm 1** Workload Modeling.

---

**Require:** $RD$  /* *RD Workload requested demand trace.* */

1: **while** true **do**
2:    $t \leftarrow 0, T_w \leftarrow 180$    /* *Initialize starting time and time window size, time in second.* */
     /* *Training Part, in current time window $T_w$.* */
3:    $X \leftarrow readLogs(T_w)$    /* *Read workload trace during window time, store it in array X.* */
4:    $CT \leftarrow t * T_w$    /* *Update current time.* */
5:    $J_{st}, T_{jt}, T_{et}, T_{net}, (R_{ij} \in R), (RE_i \in RE) \leftarrow FindStatistics(X)$    /* *Get workload statistical parameters value from log array X.* */
6:    $R_t \leftarrow$ Equation (2)
7:    $RE_t \leftarrow$ Equation (3)
8:    $RR_t \leftarrow$ Equation (4)
9:    $RD_t \leftarrow$ Equation (5)
10:    $W_t \leftarrow$ Equation (6)    /* *Using derived equations in sequence to find workload demand $W_t$.* */
11:    $\bar{x}_J, \bar{x}_T, s_J^2, s_T^2 \leftarrow$ Equation (8)
12:    $\mu_j, \mu_i, \sigma_j^2, \sigma_i^2 \leftarrow \bar{x}_J, \bar{x}_T, s_J^2, s_T^2$    /* *Set random variable values.* */
13:    $P(J_{st}(j)), P(T_{st}(i)) \leftarrow$ Equation (7)    /* *Find submitted j jobs and i tasks probability.* */
14:    $(R_{ij} = RE_i) \in R \leftarrow$ Equation (9)    /* *Find average workload request.* */
15:    $P(T_{et}) \leftarrow$ Equation (10)    /* *Find Gamma constant and probability of ended tasks.* */
16:    $P(T_{net}) \leftarrow$ Equation (11)    /* *Find probability of ended tasks that are not finished.* */
17:    $t \leftarrow t + 1$    /* *Update time index.* */
     /* *Prediction Part, next time window $T_{w+1}$.* */
18:    $J_{st}(j), T_{st}(i) \leftarrow$ Equation (12)
19:    $T_{et}, T_{net}, T_{cft} \leftarrow$ Equation (13)    /* *To predict next logs parameters values.* */
20:    $R_t \leftarrow$ Equation (2), $RE_t \leftarrow$ Equation (3), $RR_t \leftarrow$ Equation (4), $RD_t \leftarrow$ Equation (5), $W_{tp} \leftarrow$ (6)    /* *Find all next parameters, where $W_{tp}$ represents predicted workload.* */
21: **end while**

---

**Table 2.** Key Symbols Used in the Analysis and their Definitions.

| Symbol | Definition |
|---|---|
| | Workload and Data center Modeling Section Symbols |
| $W$ | total workload demands |
| $J_s$ | submitted jobs during time window $T_w$ |
| $T_s$ | submitted tasks during time window $T_w$ |
| $J_e$ | total ended jobs during time window $T_w$ |
| $T_e$ | total ended tasks during time window $T_w$ |
| $T_j$ | tasks submitted in job $j$ |
| $R$ | overall resource vector (RAM, CPU, IO, Storage and Network) demand units |
| $R_{ij}$ | requested resources vector of task $i$ in job $j$ |
| $RE_i$ | release resources vector of index $i$ |
| $t$ | absolute running time |
| $R_t$ | requested resources vector at time $t$ during time window $T_w$ |
| $J_{st}$ | number of submitted jobs at $t$ time during time window $T_w$ |
| $T_{st}$ | number of submitted tasks at $t$ time during $T_w$ |
| $T_{jt}$ | number of tasks submitted in job $j$ at time $t$ |
| $RE_t$ | total released resources at time $t$ during time window $T_w$ |
| $T_{et}$ | number of ended tasks during time window $T_w$ |
| $T_{ne}$ | number of tasks that are not finished normally and will be submitted again |
| $T_{cft}$ | number of ended tasks successfully completed during time window $T_w$ |
| $T_{net}$ | number of tasks that are not finished normally and will be considered to be resubmitted in the next time window |

**Table 2.** *Cont.*

| Symbol | Definition |
|---|---|
| $T_{ret}$ | total number of resubmitted tasks during time window $T_w$ |
| $RR_t$ | re-requested resources vector at time $t$ during time window $T_w$ |
| $RD_t$ | total demanded resources at time $t$ during time window $T_w$ |
| $CT$ | current discrete time (window counts) |
| $W_t$ | workload demands at specific time $t$ |
| $P(J_{st}(j))$ | probability of submitted job $j$ during time window $T_w$ |
| $P(T_{st}(i))$ | probability of submitted task $i$ during time window $T_w$ |
| $\mu$ | the mean of Gaussian random variable |
| $\sigma^2$ | the variance of Gaussian random variable |
| $\bar{x}$ | statistical sampled mean |
| $s^2$ | statistical sampled variance |
| $\lambda$ | Gamma distribution scale factor |
| $r$ | Gamma distribution shape factor |
| $P(T_{et})$ | probability of ended tasks during time window $T_w$ |
| $X$ | logs array for one $T_w$ |
| $W_{tp}$ | predicted workload for next time window |
| Data center Modeling Symbols | |
| $M$ | data center machine list |
| $MC$ | machine capability |
| $A$ | resources class $A = 1.00 \times MC$ |
| $B$ | resources class $B = 0.75 \times MC$ |
| $C$ | resources class $C = 0.50 \times MC$ |
| $D$ | resources class $D = 0.25 \times MC$ |
| $E$ | resources class $E = 0.00 \times MC$ |
| $NM$ | new machine state |
| $FM$ | failed machine state |
| $UM$ | upgrade machine state |
| $NM_t$ | number of new machines in new machine state during time window $T_w$ |
| $P(NM_t)$ | probability of adding $NM_t$ machines at time $t$ |
| $T_X$ | time of machine staying in the same state, $X$ means any states in a Bayesian network |
| $A_t$ | number of new machines in any state $X$, in Bayesian network |
| $\lambda_X$ | average arriving rate in Poisson and exponential random distributions of any machine state $X$ in a Bayesian network, there will be $\lambda_X$ and $X$ means any states in Bayesian network |
| $S$ | Bayesian network matrix |
| $MList$ | data center machine list capabilities |
| $MS$ | data center machines states statistics |
| $Trans$ | active scaling transition matrix |
| $PTrans$ | predicted active scaling transition matrix |

### 4.1.2. Data Center Modeling

For data center modeling, the final output is the provisioned resources *PR* that aims to match the workload demand *W*. As mentioned in Section 3, there are four machine types in the machine list *M* that are provisioned to serve the demanded resources vector *R*. Machine types are classified based on their available resource size, where the resource unit is normalized to the maximum machine resource size. The four machine types are now classified into five classes as a ratio of maximum machine capability *MC*. Class *A* machine type has a maximum resource value of $A = 1.0 \times MC$, followed by class $B = 0.75 \times MC$, class $C = 0.50 \times MC$, class $D = 0.25 \times MC$ and class $E = 0 \times MC$. Class *E* represents a machine type that has no available resources. The average requested resource vector is also normalized to *MC*, where $R_{ij} = 0.08 \times MC$. This value is considered for the optimization of the provisioned resources by selecting the optimal machine sub-list from list *M*. The data center model depends on machine states that, in turn, depend on the transition probability between classes. Google data center is provisioned gradually, as depicted in Figure 8, where demands fluctuate in shape, but the provisioned data center resources increase smoothly and gradually. This means the machine transition between classes follows a contiguous

sequence, which can be modeled using a Bayesian network. Bayesian networks use the five machine classes as states, in addition to the following three states: (a) adding a new machine *NM*, (b) removing or replacing a failed machine *FM* and (c) upgrading a machine *UM*. These three states are auxiliary states for data center capacity scaling and maintenance that are included in Bayesian networks to reflect data center machine configuration. Figure 9 presents a full diagram of the machine states in a Bayesian network and their relationships. In Definition 1, a Bayesian network must be an acyclic graph where an event sequence occurs in chronological order. To scale up the resources, each machine will transit based on the previous evidence (event already happened) in a sequential order of states. The same assumption is applied for the scaling down of resources but in the reverse sequential order. Both models of scaling types (in/out) are integrated together as Markov hidden layers architecture, as in Figure 9. The transition condition follows the conditional probability of machine states that happen as a sequence of events for each machine that is selected in both scaling types.

The Alibaba data center scaling does not have a sequential process in adding machines because there is a high fluctuation in provisioning the resources, as seen in Figures 5 and 6. On the other hand, the Google data center has smooth scaling, as shown in Figure 8. A Bayesian network model is, therefore, not a good model for Alibaba data center scaling.

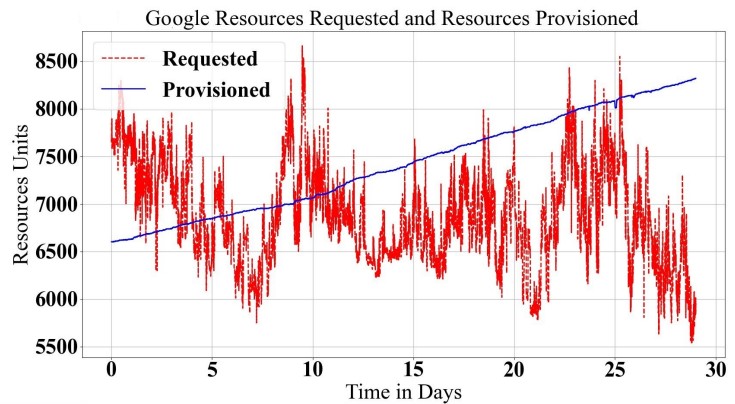

**Figure 8.** Data Center Provision Resources Shape.

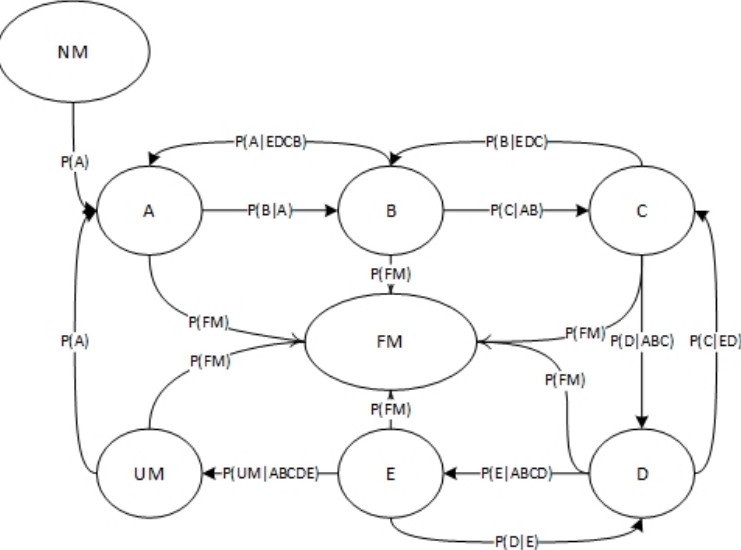

**Figure 9.** Machine States in a Bayesian Network.

**Definition 1** (Bayesian Network Definition)**.** *Let*

$$X = (X_1, \ldots, X_n)$$

*be a set of random variables. A Bayesian network is a directed acyclic graph (DAG) that specifies a joint distribution over X as a product of local conditional distributions, one for each node. Equation (14) depicts the probability distribution.*

$$P(X_1 = x_1, ..., X_n = x_n) = \prod_{i=1}^{n} P(x_i x_{parents(i)}). \tag{14}$$

The transition probability values are found using statistical analysis and probability random variable modeling. The probability of adding new machines $P(NM_t)$ in the new machine $NM$ state follows a uniform distribution over the time window, as Equation (15) depicts. $M_n$ is the average number of added machines in time window $T_w$. The new machine state $NM$ is the state representing the installation of new servers in a data center but not in production mode, where new resources are not yet available. The upgraded state $UM$ also represents machines in non-production modes that are not participating in data center operations. After adding the new servers or machines, the new machine will be in production mode starting from state $A$ (where all new and upgraded machines have maximum resources, $MU$). The probability $P(A)$ of moving new and upgraded machines to the serving mode follows a Poisson random process with an average arrival rate $\lambda_A$ in time window $T_w$. Using Amdahl's law of new machines, $A_t$ is found in state $A$ as a queuing model, as Equation (16) depicts, where $T_A$ is the serving time (time of machine staying in the same state).

$$P(NM_t) = \frac{1}{M_n}. \tag{15}$$

$$P(A) = P(NM_t) \times T_w,$$
$$A_t = \frac{\lambda_A \times T_w}{T_A}. \tag{16}$$

Based on the Bayesian network definition described in Equation (14), the probability of machine state transitions is dependent on the previous state evidence and average serving time in its states $T_A, T_B, T_C, T_D, T_E, T_{UM}, T_{NM}$. For failure machine state $FM$, time $T_{FM} = \infty$, which means no machine will get out of that state. We call it a black hole state. The transition probability between all states is found statistically by reading the real log traces for all states. The probability of failure follows an exponential random variable with an expected value of $\lambda_{FM}$, which is also found statistically. Using the aforementioned relations and equations, the data center is modeled as described in Algorithm 2. It starts by reading the logs and extracting the statistical values needed to build the Bayesian network transition matrix $S$, as in Equation (17).

$$S = \begin{bmatrix} 1 & P(BA) & 0 & 0 & 0 & 0 & 0 & P(FM) \\ P(AEDCB) & 1 & P(CAB) & 0 & 0 & 0 & 0 & P(FM) \\ 0 & P(BEDC) & 1 & P(DABC) & 0 & 0 & 0 & P(FM) \\ 0 & 0 & P(CED) & 1 & P(EABCD) & 0 & 0 & P(FM) \\ 0 & 0 & 0 & P(DE) & 1 & P(UMABCDE) & 0 & P(FM) \\ P(A) & 0 & 0 & 0 & 0 & 1 & 0 & P(FM) \\ P(A) & 0 & 0 & 0 & 0 & 0 & 1 & 0 \\ 0 & 0 & 0 & 0 & 0 & 0 & 0 & 1 \end{bmatrix} \tag{17}$$

Using statistical analysis for the transition table, we found the values of the $S$ matrix as in Table 3. The probability values are measured every time window $T_w$ during the training phase to be used in the next prediction time window. The next phase will use the pre-built Bayesian transition table of the machine list for the prediction of the data center's scaling value for the next time window.

---

**Algorithm 2** Data Center Modeling.

---

**Require:** *MList*   /* MList Data center machine list information. */

1: $t \leftarrow 0$, $T_w \leftarrow 180$, $T_{FM} \leftarrow 99999999999$   /* Initialize starting time, time window size and failed state time, time in second. */

2: **while** true **do**

   /* ***Training Part, in current time window*** $T_w$ ***/***

3:   $MC \leftarrow getMachinesCapacity(T_w, MList)$        /* Get Data center Machines capacity size from logs. */

4:   $S \leftarrow GetMachinesStatesProbability(T_w, MList)$        /* Read Data center Machines state statistical values. */

5:   $T \leftarrow GetMachinesStatesTime(T_w, MList)$        /* Find average time of states $T_A$, $T_B$, $T_C$, $T_D$, $T_E$, $T_{UM}$, $T_{NM}$ during $T_w$. Generate full matrix **T**. */

6:   $M_n, P(NM_t) \leftarrow FindStatics(MList)$        /* Find average number of added machine and their probabilities during time window $T_w$. */

7:   $MS \leftarrow [S\ T]$        /* Build data center machines states statistical tensor values. */

8:   $Trans \leftarrow TransitionActions(MS)$        /* Build transitions tracking table in Trans matrix from MS. */

   /* Find scaling values by reading transition actions, transition from state A to state UM is to provision resources, and transition from state UM to A is to de-provision resources.

9:   $S_{out} \leftarrow \sum_{i=a}^{um} Trans$        /* Find scale out $S_{out}$. */

10:  $S_{in} \leftarrow -\sum_{i=um}^{a} Trans$        /* Find scale in $S_{in}$. */

11:  $PR \leftarrow S_{out} + S_{in}$        /* Find over all provisioned value PR. */

12:  $CT \leftarrow t \times T_w$        /* Update current time. */

13:  $t \leftarrow t + 1$        /* Update time index t. */

   /* ***Prediction Part, next time window*** $T_{w+1}$ ***/***

14:  **for** item in $MList$ **do**

15:    $PTrans \leftarrow$ Equation (15), Equation (16) MS($item$)

16:  **end for**        /*Using **MS** matrix iterate over predicted next machine state using Equations (15), (16) for all states A to UM to create prediction transition matrix PTrans*/

17:  $PS_{out} \leftarrow \sum_{i=a}^{um} PTrans$

18:  $PS_{in} \leftarrow -\sum_{i=um}^{a} PTrans$

19:  $PR_t \leftarrow S_{out} + S_{in}$        /* Find predicted scale value PR*/

20: **end while**

---

**Table 3.** Bayesian Network Transition Probability Table.

| State | A | B | C | D | E | UM | NM | FM |
|:-----:|:----:|:----:|:----:|:----:|:----:|:----:|:----:|:----:|
| A | 0.12 | 0.87 | 0.00 | 0.00 | 0.00 | 0.00 | 0.00 | 0.01 |
| B | 0.16 | 0.15 | 0.68 | 0.00 | 0.00 | 0.00 | 0.00 | 0.01 |
| C | 0.00 | 0.25 | 0.14 | 0.60 | 0.00 | 0.00 | 0.00 | 0.01 |
| D | 0.00 | 0.00 | 0.39 | 0.12 | 0.48 | 0.00 | 0.00 | 0.01 |
| E | 0.00 | 0.00 | 0.00 | 0.40 | 0.10 | 0.49 | 0.00 | 0.01 |
| UM | 0.95 | 0.00 | 0.00 | 0.00 | 0.00 | 0.04 | 0.00 | 0.01 |
| NM | 0.91 | 0.00 | 0.00 | 0.00 | 0.00 | 0.00 | 0.09 | 0.00 |
| FM | 0.00 | 0.00 | 0.00 | 0.00 | 0.00 | 0.00 | 0.00 | 1.00 |

*4.2. Machine Learning Modeling Approach*

In this section, deep machine learning methods are used to build the workload and data center models. The machine learning models will simulate a data center and workload behavior to predict the workload demands and the data center's provisioned resources.

The analytical approach is compared with three types of deep machine learning methods: (1) Neural Network (NN), (2) Convolutional Neural Network (CNN) and (3) Long-Term Short Memory (LSTM) method as a type of Recurrent Neural Network (RNN). The goal of modeling is to forecast the workload demand, and the proper data center configuration needs to match the demands of a real, operational data center. In addition, it is to be used for the validation and evaluation of the proposed optimization module, which will be discussed later. The three machine learning methods model the workload and data center, considering different perspective attributes and correlations. Building the neural network with deep layers will increase the connections and operations that can handle the model nonlinear behavior complexity [43].

In the NN method, the machine learning model depends on a fully connected Neural Network using feed-forward dense layers, where the hidden layers are stacked on top of each other. Each neuron in its layer is connected to every neuron in the previous layer. Increasing the number of layers may cause an overfitting problem. However, with high-density attributes and non-convex target problems, increasing the number of layers with a proper relation design can increase the model's accuracy. With this model, no spatial information is passed. It works by interactions between layers and using many parameters to update the input weight for each layer. No special configuration is applied. The model is built as a sequence of five layers, starting with an input layer of 32 neurons for 32 attributes, then with "Dense" layers of 64, 128, 64 and 32 neurons. Finally, a one-neuron "Dense" layer is applied for regression output. All the logs are generalized and rearranged based on the predicted attributes. For example, for workload, the total demand is considered as the output attribute, and the other remaining attributes are considered as input variables. The same assumption applies to the other models.

In the CNN method, a convolution operation is applied to correlate between attributes and extract the model features, which can help model the accuracy and parameter complexity reductions. A new layer is added to the previous model after the input layer applies a convolution operation using the filter "Conv2D" after changing the input shape from a row of $(32 \times 1)$ vectors into a $(8 \times 4)$ array. The next layer is "MaxPooling2D", which works on downsampling the input representation by taking the maximum value over the convolution filter window along the feature axis. Then, a "Flatten" layer is applied that adds an extra channel dimension and output data batch size. Finally, 64, 32 and 1 "Dense" layers are applied. The convolution purpose is to preserve the spatial relationship between the log's input row vector by learning log features in small patches of the log traces.

The third method, LSTM, is a type of RNN that has a memory by using self-loops between neuron layers. These self-loops allow information to persist over time between neurons. This keeps certain sequences in neural network decision operations for the fast recognition of repeated patterns. LSTM works on a variable-length sequence, tracks long-term data dependencies, maintains information about sequence order and shares parameters across the sequence. The main concept of LSTM is using a gate to control the flow of information using four gate types: (1) forget gate, which gets rid of irrelevant information, (2) store gate, which enables the storing of relevant information from the current input, (3) update gate, which selectively stores information to be used in the update cell state and 4) output gate, which returns a filtered version of the cell state. The design of the model starts with an "LSTM" layer with 32 input neurons, followed by 32, 64 and 32 "Dense" layers. Next, 32 neurons of the "LSTM" layer, followed by 16 and 1 "Dense" layers, are applied to the model.

The Google tensorflow Python API library is used to implement the three machine's learning methods, as discussed in the next section.

### 4.3. Model Implementation and Evaluation

The proposed models are implemented using Python 3.7.8, Numpy library version 1.19.5 and Google tensorflow 2.4.1. The analytical model is built based on the Google Log traces and has been used to test the machine learning model. Google trace describes

the demands of jobs and tasks as in the workload modeling. The resource demands do not follow one type of characteristic. Two workload types are used to test the proposed modeling methods. Figures 10 and 11 show a comparison between the three machine learning models' prediction accuracy compared to the analytical method. It is clear that all methods show good prediction values with minor deviations. The analytical approach shows closer predicted values to the real trace, especially in a trace that changes quickly, such as in Figure 11. However, Figure 10 shows that ML methods are closer to a real trace. Overall, all methods show good accuracy for all types of workloads, with some variation in the prediction accuracy. This is according to logs and modeling characteristics. In non-repeated logs, it is noted that LSTM strangles rapidly predicted non-periodic changes in traces, but, on the other hand, it is a very accurate prediction scheme for similar rhythmic logs. CNN is excellent for most cases but shows a shortage in small changes predictions. NN can be more generic for all types of logs and behavior schemes without any preferable accuracy for any type of workload.

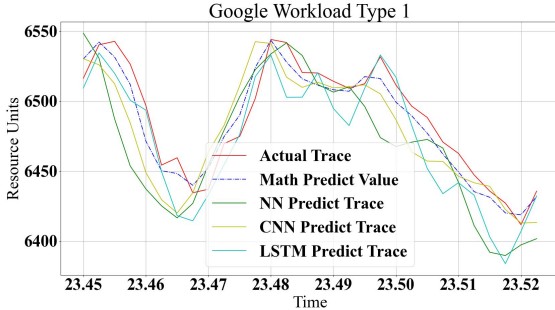

**Figure 10.** Google Workload Type 1.

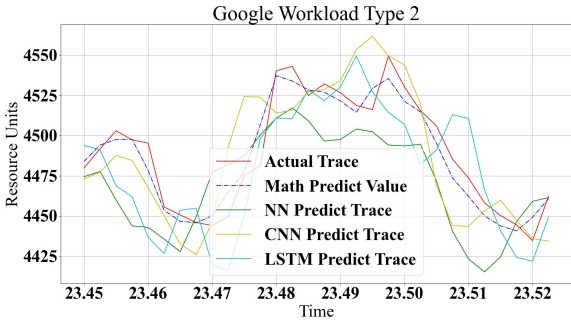

**Figure 11.** Google Workload Type 2.

For data center modeling, Figures 12 and 13 show good prediction accuracy for all kinds of models. In data centers, the values are smoother in the provisioning of resources, and log changes are lower. The ML models are very close to each other; LSTM and NN have lower accuracies in general, with small changes in log values. CNN has higher accuracy in all cases, despite this not being shown with workload traces. This is because CNN works by correlating between logs attributes, where workload attributes have a high relation argument value. Since correlation argument values are very high, the significance of the small argument changes is neglected. The small changes in log values in data centers have significant impacts since they show log characteristics. Any small changes are correlated in convolution operations and show better prediction.

For Alibaba, analytical modeling of the workload and data center traces show lower accuracy prediction. The analytical model does not match the log trace type characteristics that show intermediate prediction capabilities. On the other hand, machine learning methods provide 89% to 96% prediction accuracy. Figures 14 and 15 show the comparison between the analytical model and machine learning methods for Alibaba log prediction. In general, a summary of the differences between the prediction methods is presented in Table 4.

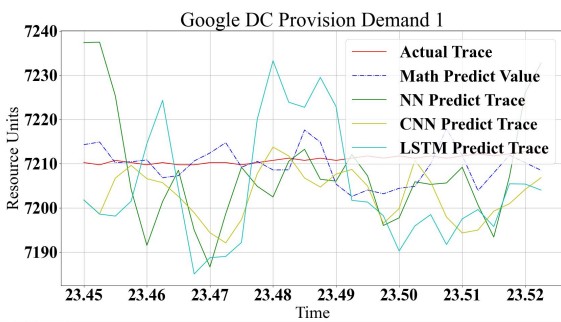

**Figure 12.** Google Data Center Provision Resources for WL1.

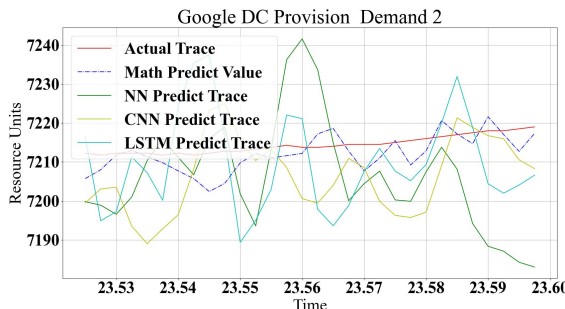

**Figure 13.** Google Data Center Provision Resources for WL2.

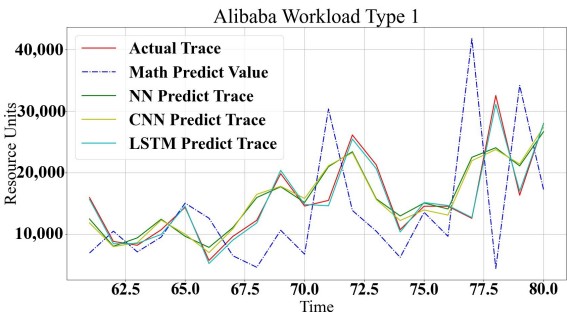

**Figure 14.** Alibaba Workload Type 1.

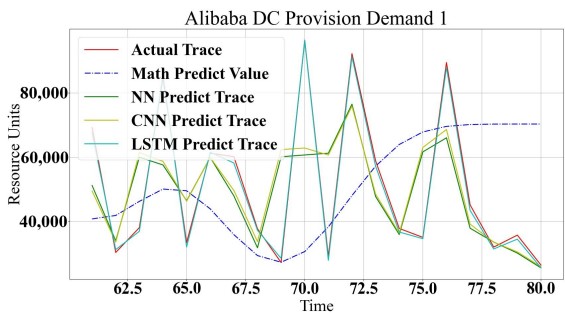

**Figure 15.** Alibaba Data Center Provision Resources for WL1.

**Table 4.** Comparison Between Machine Learning and Analytical Modeling Approaches.

| Model Type | Advantage | Disadvantage |
| --- | --- | --- |
| **Machine Learning Model** | Easy in implementation and reflecting model using automatic logs correlation<br>Generic Model<br>Automatic adaptive to changes, automatic tuning | More training time, need a high volume of logs for a more accurate model |
| **Analytical and Probability Model** | More accurate and less learning cost (time and logs size) | Nonlinearity considered, which modeled using mathematical relations and probability distribution<br>Subjective to the related driven model<br>Manual tuning, less adaptive to changes |

## 5. Elastic Scaling

In elastic scaling, the provisioned resources (*PR*) must be in the range between the Upper Bound (*UB*) and Lower Bound (*LB*), as Figure 4 depicts. In Google data centers, scaling is measured by the number of transitions between the Bayesian states in Figure 9. The number of Edges *E* in the Bayesian graph is 17. Six of these edges transition to the fail state, which are considered in the case of emergency operation. The remaining 11 transitions are for scaling operations. The metric of optimal scaling is the minimum number of transitions that gives the right provisioning resources *PR* in the fit-provisioning range $FP = UB \geq PR > LB$. The scaling action in Figure 7 is found by updating the configuration of the machine list to match the workload demands as closely as possible. Google and Alibaba trace machine tables are used to find the number of participating machines (*MCount*) in scaling action and the values of scaling that will be near the demand values, making the scale factor ($SF = \frac{PR}{RD}$) closest to 1. It is found that for Google, the number of participating machines gradually increases with respect to time. On the other hand, for Alibaba, the machines list initially starts with a high volume of machines, and later the number of machines participating decreases. A machine may go to a multiple transitions operation ($TO = \sum E$). The increase in provisioning resources in transition states *A*, *B*, *C*, *D* and *E* may return back to their own state and make the configuration more costly. The best transition happens in one direction only.

Scaling time (*ST*) is the time to find the best machine list (*MListTime*) with the least service operation time (*OP*) (time to configure machines to handle the job). In both traces, these values are not released, but we assume the operation time is fixed for all types of tasks or instances, and it can be configured in parallel. This means the scaling time can be found by adding the time slot of finding the machine's list of scaling and the service operation time (i.e., $ST = MListTime + OP$), assuming all machines are configured to serve in parallel. Based on the aforementioned characteristics for elastic scaling, we propose six metrics to evaluate the scaling: (1) scaling upper boundary violation *UB*, (2) scaling lower boundary violation *LB*, (3) scaling time *ST*, (4) number of transition operations participating in scaling *TO*, (5) number of machines participating *MCount* and (6) scaling factor *SF*.

*Results of Elastic Scaling*

Elastic scaling in this context concerns data center resource reconfiguration by minimizing rapid hardware changes and matching demands by populating an accurate machines list. By applying these metrics to Google and Alibaba scaling using the real trace, and normalizing the values to the max value, the training input of the trace is represented as a percentage. It is clear that in Figures 16 and 17, the Alibaba scaling factor is very high (6 to 7 times the demand), which means resources are being wasted. The Upper Bound *UB* value is also normalized to be a percentage. For Google trace, *SF* values are 0.85 and 1.611, which means *LB* violation is higher (0.78) in workload type 1 than (0.565) in workload type 2. Machine count *MCount* represents machine participation in provisioning for that sample. Normalized to a maximum number of available machines, *TO* represents the number of machine transitions normalized to the maximum movements in a Bayesian network. For both Alibaba and Google traces, *TO* is higher than *MCount* because one machine can jump between states more than one time. For Alibaba, most of the machines are participating in the provisioned resources from the service starting point, which increases the *ST* and *UB* factors. However, for Google, resources are added gradually, which increases the *LB* ratios and reduces *ST* costs. Further, the cloud manager needs more time to check the higher number of machines.

Table 5 summarizes the mathematical notation symbols and their definitions. In the next section, LSTM machine learning will be used to enhance these scaling metrics by optimizing elastic scaling.

**Table 5.** Key Symbols Used in the Analysis and Their Definitions.

| Symbol | Definition |
|:---:|:---|
| | Elastic Scaling Section Symbols |
| $E$ | number of edges in Bayesian graph for each transition |
| $MCount$ | number of machines participating |
| $TCount$ | number of transitions of a machine |
| $SF$ | scale factor ratio of provisioned to demand (best close to 1) |
| $TO$ | transition operation (number of transitions in Bayesian network) |
| $ST$ | scaling time |
| $MListTime$ | time to find the best machine list |
| $OP$ | service operation time (time to configure machines to handle the job) |

## 6. Machine Learning Elastic Model Optimization

Machine learning is used to reduce the data center configuration complexity (reduce machine transitions number in configuration time) and to provide the best scaling value with respect to demands. The RNN (LSTM) method is used to find an optimal provisioning operation from two perspectives (machine transition and resource provisioning). LSTM works on sequence operation and can figure out scaling sequence machine transitions. For Google traces, there is a high similarity in the machine characteristics set and the transition sequence in the same provisioning machines list. The training model is derived from the logs, finding the number of machines $MList$ that participated in the provided time window $T_w$. During that time window, finding the number of transitions $TCount$ and provisioned value $PR$ are new evaluation metrics. With this mapping, a list of configuration cases is generated to provide the same resource values. The supporting factors indicate that the best case happens when the provisioned value ($PV$) is close to workload requested demand ($RD$) when the best match scale factor $SF$ ratio is equal to 1 ($SF = 1$), and the best count ratio is close to 1 ($BC = 1/TCount$). With this normalization, the training values that must be chosen to perfect the configuration ($PFC$) are when $PFC = SF/BC \cong 1$. The LSTM model feeds the stages of Forget, Store, Update and Output as follows; if a new, perfect configuration $PFC$ value is closer to 1 than the previous, old training values, then the current machine list is discarded and a new training argument for the new machine list is generated. ML updating network parameters of neural network layers are accomplished using the new arguments to the next layers, and so on. The machine learning model here is used as a classifier, which can select the most machine list classes with a minimum transition number that can match the demands. The schema of the training set is listed in Table 6. It is clear that one provisioned value will have many configuration sets, $MCount$ in $MList$ and this many configuration sset to one provisioned value relationship is subjected to machine participation in configuration.

*Results of Elastic Optimization*

ML is used as a generic optimizer module, which targets the machine count as a minimum of the provisioned resource matching demand. In this approach, we follow a labeling scheme for data center configuration similar to Genetic Algorithm chromosome populations but with prior knowledge from existing configurations. The configuration might be rejected if one machine is not available in the list by checking the machine availability flag in the machine array $MListFlag$. With this, a dataset is generated for the data center configuration based on Table 6 to be used in the LSTM machine learning model to train it for optimizing elastic scaling in conjugation with the prediction model. The proposed analytical model is used as a model simulation for the validation and evaluation of the optimization of elastic scaling in a real data center environment. Figures 18 and 19 show a good scaling factor ratio with lower machine transitions and counts using our proposed model compared to the original data center scaling in Figures 16 and 17.

**Table 6.** Schema of Optimized Elastic Scaling.

| Attribute Names |
| --- |
| Time |
| Requested Demand (*RD*) |
| Machine participating count *MCount* from (*MList*) |
| Transitions count (*TCount*) out of all total of transition operations *TO* |
| Machine available flag array *MListFlag* |
| Provisioned value (*PV*) |
| Best match ($BM = PV/RD$) |
| Best count ($BC = 1/TCount$) |
| Perfect Configuration $PFC = BM/BC \cong 1$ |

The model is built using total workload demands as the argument input and the output matrix of the machine list and each machine's transitions, finding the best match to demand with a minimum number of machines transitions. Labeling the machine list by the classifier, which compares the actual trace vectors of scaling configuration and the machine learning-optimized configuration setup. The number of configuration classes is found by dividing the maximum number of machines participating during the whole trace time by 5 to find scaling categories, where the five *PFC* interval values are [(0.0,0.2), [0.2,0.4), [0.4,0.6), [0.6,0.8) and [0.8,1.0)].

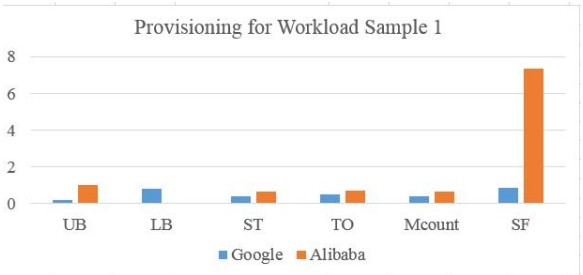

**Figure 16.** Scaling Ratios for Google and Alibaba Workload Type 1.

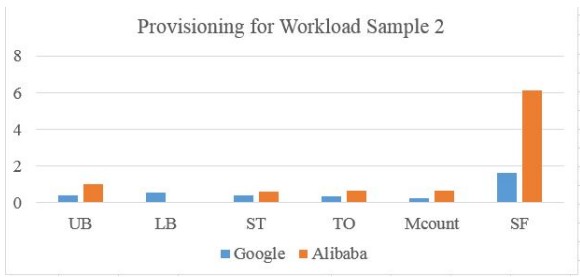

**Figure 17.** Scaling Ratios for Google and Alibaba Workload Type 2.

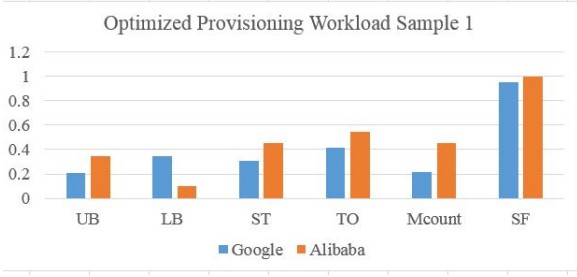

**Figure 18.** Optimized Scaling Ratios for Google and Alibaba Workload Type 1.

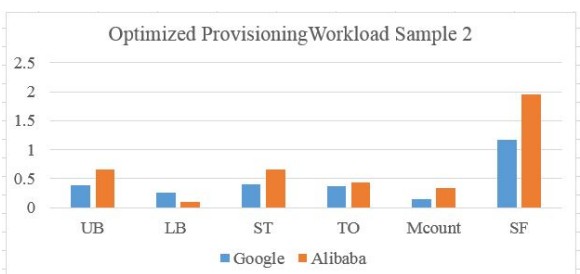

**Figure 19.** Optimized Scaling Ratios for Google and Alibaba Workload Type 2.

## 7. Conclusions

This work reduces the cloud data center configuration cost in three ways. First, reduce the running cost by choosing the best resource scaling configuration set that matches the workload demands. Second, reducing the configuration time and complexity because configuration sets are predefined and clustered, which gives a fast response and accurate decisions to the cloud data center, which further reduces power consumption and hardware stress lifetime. Third, reducing the customer usage cost by provisioning the exactly needed resources as an accurate elastic scheme. Our future work is to integrate the proposed ML models with our cloud management system developed using a micro-service architecture pattern [44].

**Author Contributions:** The authors of this work contributed equally whereas T.D. works under the supervision of A.A. The T.D. work on math derivation, design, modelling, and implementation. A.A revised all the work and keep supervision of the work. All authors have read and agreed to the published version of the manuscript.

**Funding:** This research was funded by Mitacs Accelerate program in collaboration with Cistech.

**Data Availability Statement:** We used Google [45] and Alibaba [41] data-sets are available online.

**Acknowledgments:** This work was supported by the Natural Sciences and Engineering Research Council and Mitacs Accelerate program in collaboration with Cistech.

**Conflicts of Interest:** The authors declare no conflict of interest.

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
