# Peer review of "Cloud Workload and Data Center Analytical Modeling and Optimization Using Deep Machine Learning"

_2673-8732, doi:10.3390/network2040037_

Round 1

Reviewer 1 Report

This a timely reasearch paper, which discusses the application of ML techniques in cloud management. I like the author's choice of trace from two representatitive cloud providers and conduct the comparative evaluation on them. The conclusion is insightful based on the analysis, which motivates the future design of new auto-scaling algorithms for cost-effectiveness management.

I hope  the authors can add some introduction of reinforcement learning related techniques in the background/related work section. In recent years, RL is  also popular in this area of AI-based cloud operation. The ML techniques involved in this paper are all classic ones. It would be great to introduce some recent fresh flavor into the paper.

Author Response

Dear respected reviewer,

Thank you very much for your valuable comments, please find our reply attached.

Sincerely,

Authors   

Reviewer 2 Report

This paper outlines some approaches to predicting workload and datacenter operations.

The topic considered by the authors is very hot in the cloud field and the solutions already proposed are countless. The authors take this into account by presenting a rich section of related work.

Then they propose a rich experimental campaign underlying their proposal. This experimental campaign allows them to identify three possible alternative approaches based on machine learning and to choose one of them as optimal.

In my opinion, the paper is suitable for publication but has several typos that should be removed (e.g., "detacenter" in the abstract).

Author Response

(The authors gave the same response as above.)

Reviewer 3 Report

This paper presents different models to predict workload in datacenters and to optimize resources dedicated to serve the computation requests. To this end, logs traces from Google and Alibaba datacenters are used to train 3 different kind of Neural Networks showing

In general I like the paper and the idea of using ML to optimize datacenter resource utilization and consumption. The main concern I have with this paper is that I do not fully understand why the datacenter model is not taking any advantage of the predicted workload. Since the goal is to achieve an automatic and optimized elastic scaling of the available resources, it seems  like a good idea to use these predictions, given that they seem accurate enough. This might have something to do with the datasets, but I am not that familiar with them, so I cannot really judge that.

Thus, and related with the datasets, it would be nice if they were further described in the paper, how many traces they have, what other information can you get from them besides workload and resources used (if any) and so on. Also, is there any reference to the Google datasets?

The paper mentions that the Google traces have a bell shape, which would match to a Gaussian distribution, but was the model tested? if so what was the error? If not, and in both cases, I think it would be interesting to estimate how well the actual number of arriving jobs fits the proposed distribution.

Most of the results are presented with figures, and by looking and analyzing them I cannot really tell the accuracy of the models. Further, the analysis of the results is purely qualitative like “they provide good prediction accuracy”. In this case I would be quite interesting to have actual numbers about accuracy and errors in the estimations.

As for the related work section I have a couple of comments:

Line 98 states that “most of the optimization methods are subjective to limited number of case scenarios”, does it refer to all previous works? If so, it seems that this should not be the case for the presented paper, and I do not really see how this paper deals with that.

I am not aware of the details of [18], could you elaborate more why it might violate elastic condition? Otherwise it is hard to understand the criticism.

Finally, Figure 7 is somehow confusing, it is unclear if the same entity that resubmits unfinished tasks is the same as the one setting up the configuration.

Author Response

(The authors gave the same response as above.)

Reviewer 4 Report

This paper talked about three methods of machine learning used and compared with the analytical approach to model workload and datacenter actions. The presentation is understandable in general. The simulation results are included to illustrate the model.

I think the research problem is well motivated and the technical details seem solid.

However, I am not satisfied with the contribution of the article. The main concern is the lack of comparison with other models. 

Some comments are as follows:

1. The English language should be improved.

2. Introduction: Very limited in the first section and quickly moved ahead with the problem statement research motivation. Here, you need to provide detailed introduction based on the your research with clear context of what you need to do. Make the problem statement and the research statement concise, as you describe steps and background in research motivation. Further, in the same section you have a research contribution. You need to clearly align all these concepts differently.

3. In related work, up-to-date cloud system models should be discussed.

4. I tried to find the results section, but did not find it. Lack of discussion section. Directly, a jump to summary. No recommendations: no future work, no managerial implications, no details for replication of study.

11. Very few references. There are about more than 1000+ articles specifically on this topic since 2020. But the authors found only a few and many of them are not relevant.

Author Response

(The authors gave the same response as above.)
